

# The effect of dephasing and spin-lattice relaxation during the switching processes in quantum antiferromagnets

**Asliddin Khudoyberdiev**[⋆] **and Götz S. Uhrig**[†]

Condensed Matter Theory, TU Dortmund University,
Otto-Hahn-Straße 4, 44227 Dortmund, Germany

⋆ asliddin.khudoyberdiev@tu-dortmund.de , † goetz.uhrig@tu-dortmund.de

## Abstract

The control of antiferromagnetic order can pave the way to large storage capacity as well as fast manipulation of stored data. Here achieving a steady-state of sublattice magnetization after switching is crucial to prevent loss of stored data. The present theoretical approach aims to obtain instantaneous stable states of the order after reorienting the Néel vector in open quantum antiferromagnets using time-dependent Schwinger boson mean-field theory. The Lindblad formalism is employed to couple the system to the environment. The quantum theoretical approach comprises differences in the effects of dephasing, originating from destructive interference of different wave vectors, and spin-lattice relaxation. We show that the spin-lattice relaxation results in an exponentially fast convergence to the steady-state after full ultrafast switching.

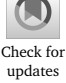

# 1  Introduction

The ultrafast spin-dynamics and the absence of stray fields in antiferromagnets propose their application in spintronics devices [1]. The usage of antiferromagnets for data storage has the potential to improve the information processing time scale by a factor of 1000 and significantly enlarge storage capacity [2–4]. However, the existence of intrinsic terahertz (THz) frequencies does not automatically guarantee ultrafast switching due to the difficulty of efficiently and fully controlling the magnetization at the microscopic level. Studies are ongoing to extensively explore the fast dynamics in antiferromagnets, with the primary focus on the efficient manipulation and control of their magnetic state in the THz regime [5–7].

The absence of net magnetization in antiferromagnets poses a significant challenge in the efficient readout of the direction of the Néel vector. The electrical readout methods based on anisotropic magnetoresistance and the planar Hall effect are well-established techniques [8,9]. There are also techniques based on optical means to readout the Néel vector [4, 10]. Furthermore, an extremely strong exchange coupling between the Néel vector of the metallic antiferromagnet $Mn_2Au$ and the magnetization of the ferromagnetic Py layer enables the electric detection of the Néel vector orientation via standard techniques used for ferromagnetic thin films [11]. The deflection of the Néel vector by 30° has been experimentally obtained in $Mn_2Au$ at ultrafast time scale through the action of so-called spin-orbit torques, and the results are consistent with the micromagnetic model [12]. Moreover, atomistic spin dynamics simulations predict the possibility for the exchange-enhanced switching of the Néel vector by 90° and 180° using novel laser optical torques [13]. The classical Landau-Lifshitz-Gilbert equations also confirmed the stable current-induced precession of the Néel vector [14, 15].

A quantum approach has been developed to study the switching in quantum antiferromagnets driven by external magnetic fields, based on time-dependent Schwinger boson mean-field theory [16–18]. This approach has demonstrated that control of the Néel vector can be achieved through the application of strong uniform fields [16, 17]. Moreover, staggered magnetic fields in neighboring sublattices generate exchange field enhancement. As a result, switching occurs under significantly lower fields [18], because the internal exchange fields are several orders of magnitude larger than the driving external fields and assist to reorient the order [8, 12, 15, 19]. Additionally, despite the quantum system being closed, the dynamics of sublattice magnetization after switching is not coherent, but a slow decay of the oscillations is observed. This phenomenon has been claimed to be a dephasing effect caused by the numerous different frequency modes in the system [16–18]. The effect of dissipation has not yet been considered, where the spin system can exchange an energy with its environment, e.g. with a thermal bath such as generated by all lattice vibrations.

For the practical application of this technology, the magnetic state needs to be robust against external noise effects to keep stored information safe and secure. The effect of the environment is of particular significance and inevitable, especially if one aims at reaching the switched coherent stationary state of the system quickly. For large magnetic samples, the dynamics of the magnetization can be well described by quantum excitations involving numerous lattice sites. Describing the dynamics of open many-body quantum system is a substantial challenge for modern physics. A powerful tool to analyze dissipative many-body quantum

system is the Lindblad approach [20]. Therefore, we employ quantum theory to analyze sublattice magnetization switching processes in quantum antiferromagnets, taking into account the spin-lattice relaxations derived from the Lindblad formalism. To this end, we use the time-dependent Schwinger boson mean-field theory at finite temperature, and the magnetization control is obtained directly via an external magnetic field.

The objective of this study is to obtain coherent steady-state in quantum antiferromagnets after switching of the sublattice magnetization. The dissipation can speed up the decay of oscillations in magnetization after switching, as it drives the system towards a new ground state. Consequently, we extend the investigation of exchange-enhanced switching in quantum antiferromagnets [18] by incorporating environmental effects in the framework of open quantum systems (Figure 1). To the best of our knowledge, time-dependent Schwinger boson mean-field theory is the only quantum approach that has been employed to analyze the switching of quantum many-body system, and we adhere to this approach. The exploration of other alternative methods lies beyond the scope of the present work. Here, we consider the quantum antiferromagnetic square lattice as an exemplary model. It can be extended to other lattice structure, especially to 3d lattices, in future works.

The realization of seemingly impossible local fields that alternate their orientation between the sublattices of the antiferromagnet can be envisaged by utilizing global field which act locally different on each sublattice due to the anisotropy of the system [8, 12]. For example, the theoretical approach [19] and experimental observations [9, 21] show that globally applied current-induced spin orbit torques can exhibit a Néel-type character, manifesting itself as locally alternating fields between the two sublattices. Furthermore, the $g$ tensor can be anisotropic due to the large spin-orbit coupling so that it differs between the two sublattices, i.e., $\underline{\underline{g}}|_A \neq \underline{\underline{g}}|_B$. Consequently, a globally applied external magnetic field generates locally alternating components. Thus, the local control of sublattice magnetization using global fields is a key prerequisite for making antiferromagnetic spintronics feasible [3].

The paper is organized as follows. In Section 2, we provide a theoretical model for switching and define the equilibrium state of the system. The switching in closed quantum antiferromagnets is discussed in Section 3 to show the effect of dephasing only. In Section 4, we investigate an open quantum antiferromagnet and the role of dissipation on magnetization switching. Section 5 is devoted to the conclusion, together with an outlook of the future directions envisioned for the current research.

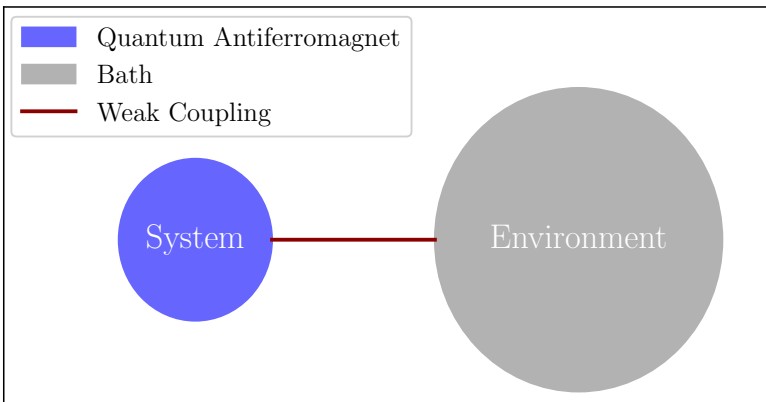

Figure 1: The illustration of the system that is weakly coupled to an environment, e.g., lattice vibrations and hence spin-phonon interactions are taken into account.

# 2 Time-dependent Schwinger boson mean-field theory

We use a spin-1/2 anisotropic Heisenberg model with nearest-neighbor interactions on a square lattice quantum antiferromagnet and its Hamiltonian is given by

$$\mathcal{H}_0 = J \sum_{\langle i,j \rangle} \left[ \frac{\chi}{2} (S_i^+ S_j^- + S_i^- S_j^+) + S_i^z S_j^z \right], \tag{1}$$

where $\chi = J_{xy}/J_z \in [0,1]$ is an anisotropy parameter and $J$ is the coupling constant.[1]

An external magnetic field is included as a Zeeman term to manipulate the system. To this end, it has been demonstrated that the staggering fields can reorient the Neél vector efficiently due to intrinsic induced exchange enhancement [15, 18]. In this context, we aim for the switching at low fields with the corresponding term in the Hamiltonian reading

$$\mathcal{H}_{\mathrm{m}} = -\mathbf{h} \cdot \sum_i (-1)^i \mathbf{S}_i, \tag{2}$$

where index $i$ runs over all the lattice cites in the system. The switching implies strong fluctuations which drive the system far from the equilibrium. The Schwinger boson representation is chosen as an appropriate approach to describe the switching process [16, 17], because Schwinger bosons can capture magnetic orders with arbitrary orientation [22, 23]. In this representation, the spin operators are expressed in terms of two bosonic species, which together are capable of capturing any possible orientation of the spin, as

$$S_i^+ = a_i^\dagger b_i, \quad S_i^- = b_i^\dagger a_i, \quad S_i^z = \frac{1}{2} \left( a_i^\dagger a_i - b_i^\dagger b_i \right), \tag{3}$$

including the constraint on the bosonic number

$$a_i^\dagger a_i + b_i^\dagger b_i = 2S. \tag{4}$$

The above local constraint restricts the Fock space of the bosons to the meaningful physical subspace of spin $S$. It is important to note that the elementary excitations differ here from the conventional magnons in spin-wave theory, as the Schwinger boson Hamiltonian is quartic in bosonic operators, thereby the interactions are incorporated into the theory. The sublattice magnetization can be calculated as

$$m = \langle S_i^z \rangle = \frac{1}{2} \left( \langle a_i^\dagger a_i \rangle - \langle b_i^\dagger b_i \rangle \right). \tag{5}$$

It can be seen that the orientation of sublattice magnetization can be controlled by controlling the bosonic occupation number of the lattice site. For this purpose, we first determine a proper equilibrium state of the system, i.e., the initial mean-occupation number of bosonic species on the sublattices. Subsequently, the external switching field is applied that can induce non-equilibrium dynamics by changing the boson numbers in the system. The equations for the mean occupation number of bosons will be constructed using the Heisenberg equation of motion.

## 2.1 An equilibrium state of the system

We start by rotating the spins of one sublattice by 180° about $S_j^{y\,2}$ to obtain a uniform description of the system. This is a canonical transformation and it preserves the constraint in

---

[1]Throughout this work, $J$ is chosen as the unit of energy, and is henceforth set to unity.

[2]Rotation in one sublattice with index $j$ in Eq. (1) which applies for Schwinger bosons $a_j \to -b_j$, $b_j \to a_j$, where index $j$ belongs to only one sublattice type in the system, e.g., only sites with spin down.

Eq. (4) [22, 24]. Consequently, $S_j^x$ undergoes a sign change, and the lattice sites experience the alternating external field in Eq. (2) along the $x$ direction as though it were uniform. In parallel, this considerably simplifies our analyses because the staggered field in Eq. (2) becomes homogeneous. Next, we replace the spin operators in Eq. (1) by Schwinger bosons from Eq. (3) and a bilinear Hamiltonian results after the mean-field approximation [16]. The mean-field Hamiltonian in momentum space becomes

$$\mathcal{H}_0^{\text{MF}} = E_0 - \frac{1}{2}\sum_{\mathbf{k}}\gamma_{\mathbf{k}}\big(C_-a_{\mathbf{k}}^{\dagger}a_{-\mathbf{k}}^{\dagger} + C_+b_{\mathbf{k}}^{\dagger}b_{-\mathbf{k}}^{\dagger} + C_-^*a_{\mathbf{k}}a_{-\mathbf{k}} + C_+^*b_{\mathbf{k}}b_{-\mathbf{k}}\big) + \lambda\sum_{\mathbf{k}}\big(a_{\mathbf{k}}^{\dagger}a_{\mathbf{k}} + b_{\mathbf{k}}^{\dagger}b_{\mathbf{k}}\big), \quad \text{(6a)}$$

$$\mathcal{H}_{\text{m}}^{\text{MF}} = -\frac{h}{2}\sum_{\mathbf{k}}\big(a_{\mathbf{k}}^{\dagger}b_{\mathbf{k}} + b_{\mathbf{k}}^{\dagger}a_{\mathbf{k}}\big), \quad \text{(6b)}$$

$$\mathcal{H}_{\text{MF}} = \mathcal{H}_0^{\text{MF}} + \mathcal{H}_{\text{m}}^{\text{MF}}, \quad \text{(6c)}$$

where $A := \langle a_i a_j + b_i b_j\rangle$, $B := \langle a_i a_j - b_i b_j\rangle$ and $C_\pm := A(1+\chi)\mp B(1-\chi)$ with $A, B \in \mathbb{C}$. $E_0$ is the ground state energy, and the wave factor $\gamma_{\mathbf{k}}$ includes the wave vector as $\gamma_{\mathbf{k}} = \frac{1}{z}\sum_{\boldsymbol{\delta}}e^{i\mathbf{k}\cdot\boldsymbol{\delta}}$. The Lagrange term with the Lagrange parameter $\lambda$ is included in the Hamiltonian to restrict the number of bosons per site and ensure that the constraint in Eq. (4) is fulfilled on average. One should note that as a result of the sublattice rotation, the bond operators become effectively translationally invariant, namely $A = \langle A_{ij}\rangle$, $B = \langle B_{ij}\rangle$ and $A^* = \langle A_{ij}^{\dagger}\rangle$, $B^* = \langle B_{ij}^{\dagger}\rangle$.

The above mean-field Hamiltonian is diagonalized by introducing bosonic Bogoliubov operators

$$\alpha_{\mathbf{k}}^{\dagger} = \cosh(\theta_{\mathbf{k}}^a)a_{\mathbf{k}}^{\dagger} - e^{-i\phi_{\mathbf{k}}^a}\sinh(\theta_{\mathbf{k}}^a)a_{-\mathbf{k}}, \quad \text{(7a)}$$

$$\beta_{\mathbf{k}}^{\dagger} = \cosh(\theta_{\mathbf{k}}^b)b_{\mathbf{k}}^{\dagger} - e^{-i\phi_{\mathbf{k}}^b}\sinh(\theta_{\mathbf{k}}^a)b_{-\mathbf{k}}. \quad \text{(7b)}$$

The Bogoliubov angles $\theta_{\mathbf{k}}^{a,b}$ necessary for the diagonalization condition can be represented as

$$\tanh 2\theta_{\mathbf{k}}^a = \frac{C_-\gamma_{\mathbf{k}}e^{-i\phi_{\mathbf{k}}^a}}{\lambda}, \qquad \tanh 2\theta_{\mathbf{k}}^b = \frac{C_+\gamma_{\mathbf{k}}e^{-i\phi_{\mathbf{k}}^b}}{\lambda}. \quad \text{(8)}$$

Then, the dispersion relations read

$$\omega_{\mathbf{k}}^{\pm} = \sqrt{\lambda^2 - |C_\pm|^2\gamma_{\mathbf{k}}^2}, \quad \text{(9)}$$

where $\omega_{\mathbf{k}}^-$ and $\omega_{\mathbf{k}}^+$ correspond to the $\alpha_{\mathbf{k}}$ and $\beta_{\mathbf{k}}$ bosons, respectively. The spin gap

$$\Delta := \Delta^+ - \Delta^- = \omega_{\mathbf{k}}^+|_{\mathbf{k}=0} - \omega_{\mathbf{k}}^-|_{\mathbf{k}=0}, \quad \text{(10)}$$

functions as a energy barrier and is of particular relevance to obtain the switching [16, 17] because the system requires an energy input sufficient to surpass the potential barrier associated with the transition between opposite antiferromagnetic orders. Overcoming this barrier is achieved through the applied switching field in the Hamiltonian (6b), and we assume that the field is turned on at time $t = 0$ and switched off at $t = 10J^{-1}$.

From the diagonalization conditions in Eq. (8), one can construct self-consistent equations to find the mean-field parameters and the Lagrange parameter as

$$\begin{cases} A = \langle a_i a_j\rangle + \langle b_i b_j\rangle = \frac{1}{N}\sum_{\mathbf{k}}\gamma_{\mathbf{k}}\big(\langle a_{\mathbf{k}}a_{-\mathbf{k}}\rangle + \langle b_{\mathbf{k}}b_{-\mathbf{k}}\rangle\big), \\ B = \langle a_i a_j\rangle - \langle b_i b_j\rangle = \frac{1}{N}\sum_{\mathbf{k}}\gamma_{\mathbf{k}}\big(\langle a_{\mathbf{k}}a_{-\mathbf{k}}\rangle - \langle b_{\mathbf{k}}b_{-\mathbf{k}}\rangle\big), \\ 2S = \langle a_i^{\dagger}a_i\rangle + \langle b_i^{\dagger}b_i\rangle = \frac{1}{N}\sum_{\mathbf{k}}\big(\langle a_{\mathbf{k}}^{\dagger}a_{\mathbf{k}}\rangle + \langle b_{\mathbf{k}}^{\dagger}b_{\mathbf{k}}\rangle\big), \end{cases} \quad \text{(11)}$$

where

$$\begin{cases} \langle a_{\mathbf{k}}^{\dagger} a_{\mathbf{k}} \rangle = \frac{\lambda}{\omega_{\mathbf{k}}^{-}} \left( n(\omega_{\mathbf{k}}^{-}) + \frac{1}{2} \right) - \frac{1}{2}, \\ \langle b_{\mathbf{k}}^{\dagger} b_{\mathbf{k}} \rangle = \frac{\lambda}{\omega_{\mathbf{k}}^{+}} \left( n(\omega_{\mathbf{k}}^{+}) + \frac{1}{2} \right) - \frac{1}{2}, \\ \langle a_{\mathbf{k}} a_{-\mathbf{k}} \rangle = \frac{C_{-}\gamma_{\mathbf{k}}}{\omega_{\mathbf{k}}^{-}} \left( n(\omega_{\mathbf{k}}^{-}) + \frac{1}{2} \right), \\ \langle b_{\mathbf{k}} b_{-\mathbf{k}} \rangle = \frac{C_{+}\gamma_{\mathbf{k}}}{\omega_{\mathbf{k}}^{+}} \left( n(\omega_{\mathbf{k}}^{+}) + \frac{1}{2} \right), \end{cases} \tag{12}$$

with $n(\omega_{\mathbf{k}}^{\pm}) = \left( \exp\left( \beta \omega_{\mathbf{k}}^{\pm} \right) - 1 \right)^{-1}$ being the Bose distribution function. These equations complete the system's initialization for finding the proper initial state given by the thermal equilibrium in the mean-field description.

## 3 Exchange-enhanced switching in closed systems including dephasing effect

We construct a closed set of differential equations using Heisenberg's equations of motion to analyze the non-equilibrium of the system under applied pulses. The equations read

$$\begin{cases} \partial_t \langle a_{\mathbf{k}}^{\dagger} a_{\mathbf{k}} \rangle = -i\gamma_{\mathbf{k}} \big( C_{-}^{*} \langle a_{\mathbf{k}} a_{-\mathbf{k}} \rangle - C_{-} \langle a_{\mathbf{k}}^{\dagger} a_{-\mathbf{k}}^{\dagger} \rangle \big) + i\frac{h}{2} \big( \langle a_{\mathbf{k}}^{\dagger} b_{\mathbf{k}} \rangle - \langle b_{\mathbf{k}}^{\dagger} a_{\mathbf{k}} \rangle \big), \\ \partial_t \langle b_{\mathbf{k}}^{\dagger} b_{\mathbf{k}} \rangle = -i\gamma_{\mathbf{k}} \big( C_{+}^{*} \langle b_{\mathbf{k}} b_{-\mathbf{k}} \rangle - C_{+} \langle b_{\mathbf{k}}^{\dagger} b_{-\mathbf{k}}^{\dagger} \rangle \big) - i\frac{h}{2} \big( \langle a_{\mathbf{k}}^{\dagger} b_{\mathbf{k}} \rangle - \langle b_{\mathbf{k}}^{\dagger} a_{\mathbf{k}} \rangle \big), \\ \partial_t \langle a_{\mathbf{k}} a_{-\mathbf{k}} \rangle = i\gamma_{\mathbf{k}} C_{-} (2\langle a_{\mathbf{k}}^{\dagger} a_{\mathbf{k}} \rangle + 1) - 2\lambda i \langle a_{\mathbf{k}} a_{-\mathbf{k}} \rangle + i h \langle a_{\mathbf{k}} b_{-\mathbf{k}} \rangle, \\ \partial_t \langle b_{\mathbf{k}} b_{-\mathbf{k}} \rangle = i\gamma_{\mathbf{k}} C_{+} (2\langle b_{\mathbf{k}}^{\dagger} b_{\mathbf{k}} \rangle + 1) - 2\lambda i \langle b_{\mathbf{k}} b_{-\mathbf{k}} \rangle + i h \langle a_{\mathbf{k}} b_{-\mathbf{k}} \rangle, \\ \partial_t \langle a_{\mathbf{k}}^{\dagger} b_{\mathbf{k}} \rangle = -i\gamma_{\mathbf{k}} \big( C_{-}^{*} \langle a_{\mathbf{k}} b_{-\mathbf{k}} \rangle - C_{+} \langle a_{\mathbf{k}}^{\dagger} b_{-\mathbf{k}}^{\dagger} \rangle \big) - i\frac{h}{2} \big( \langle b_{\mathbf{k}}^{\dagger} b_{\mathbf{k}} \rangle - \langle a_{\mathbf{k}}^{\dagger} a_{\mathbf{k}} \rangle \big), \\ \partial_t \langle a_{\mathbf{k}} b_{-\mathbf{k}} \rangle = i\gamma_{\mathbf{k}} \big( C_{-} \langle a_{\mathbf{k}}^{\dagger} b_{\mathbf{k}} \rangle + C_{+} \langle b_{\mathbf{k}}^{\dagger} a_{\mathbf{k}} \rangle \big) - 2\lambda i \langle a_{\mathbf{k}} b_{-\mathbf{k}} \rangle + i\frac{h}{2} \big( \langle a_{\mathbf{k}} a_{-\mathbf{k}} \rangle + \langle b_{\mathbf{k}} b_{-\mathbf{k}} \rangle \big). \end{cases} \tag{13}$$

The above equations are solved for each momentum $\mathbf{k}$ in the first Brillouin zone using the Boost Odeint library. Now, the time evolution of spin expectation values can be calculated as

$$\begin{cases} \langle S^x \rangle = \frac{1}{N} \sum_{\mathbf{k}} \gamma_{\mathbf{k}} \Re \langle a_{\mathbf{k}}^{\dagger} b_{\mathbf{k}} \rangle, \\ \langle S^y \rangle = \frac{1}{N} \sum_{\mathbf{k}} \gamma_{\mathbf{k}} \Im \langle a_{\mathbf{k}}^{\dagger} b_{\mathbf{k}} \rangle, \\ m = \langle S^z \rangle = \frac{1}{2N} \sum_{\mathbf{k}} \left( \langle a_{\mathbf{k}}^{\dagger} a_{\mathbf{k}} \rangle - \langle b_{\mathbf{k}}^{\dagger} b_{\mathbf{k}} \rangle \right). \end{cases} \tag{14}$$

The spin gap in Eq. (10) primarily controls the stiffness of the magnetization, meaning that the system requires some minimum external energy for switching to overcome the potential barrier. The threshold value of the external uniform switching field corresponds closely to the spin gap in a lattice [16,17]. Reorientation of the order can be achieved under fairly low external staggered fields due to exchange enhancement [18]. Additionally, our findings indicated that dephasing, caused by the destructive interference of many modes at different frequencies in a large closed system, leads to a temporal slow decay of sublattice magnetization oscillations after switching [17, 18]. Although the main aim of the present work is to distinguish the effect of dephasing and relaxation, we start by presenting the results of efficient switching without relaxation to highlight once more the exchange enhancement and dephasing effect. The next section will be dedicated to relaxation.

Figure 2(a) shows the dynamics of the spin expectation values obtained from Eqs. (14) for $\chi = 0.9$. The expectation value $\langle S^z \rangle$, i.e., the sublattice magnetization $m$ exhibits the switching behavior accompanied by the dephasing effect after switching. The dephasing effect slows down the oscillations after full switching, and the system attempts to reach a steady

state. The dynamics of $\langle S^x \rangle$ shows also Larmor precession around the field with a decrease in amplitude. Here we see the exchange-enhanced switching as follows; the Schwinger boson mean-field calculations show that the spin gap for a square lattice at anisotropy $\chi = 0.9$ has a value $\approx 0.86J$. Thus, one can estimate that a field of similar size is necessary for switching. However, the applied staggered field is $h = 0.08J$ in Figure 2(a) which is much lower than the potential barrier. Thus, the system benefits from strong internal exchange fields to reorient the magnetization.

Figure 2(b) illustrates the processes of exchange enhancement during the switching. Firstly, antiparallel spins in two neighboring sublattices cant slightly in different directions from their antiparallel equilibrium state because of the applied staggered field (red arrows) resulting in a magnetic moment (black arrow). Due to the exchange coupling the spins rotate about the induced magnetic moment. In other words, the Néel vector rotates around this internal strong exchange field and switching occurs. During the switching interval, the $y$ component of the spin is positive. Afterwards, it displays a small oscillating behavior. Indeed, the dynamics of $\langle S^y \rangle$ in Figure 2(a) confirms our claim of exchange enhancement. It shows small canting until the system encounters full switching (around $t = 12.5J^{-1}$) and then displays very narrow oscillating behavior around the $y$ axis.

However, one can see that a decrease in the oscillations due to the dephasing alone is insufficient to achieve the static behavior of magnetization after switching. This is not very promising for practical applications because data processing requires fast relaxation of the switched magnetic state, which guarantees stored data safety and to avoid unintentional back switching. Having exchange-enhancement and dephasing in mind, our objective is to include an additional effect of the environment on the switching process in our operator formalism approach. To this end, we propose implementing Lindblad dissipators while studying the switching in quantum antiferromagnets to improve the static behavior after switching.

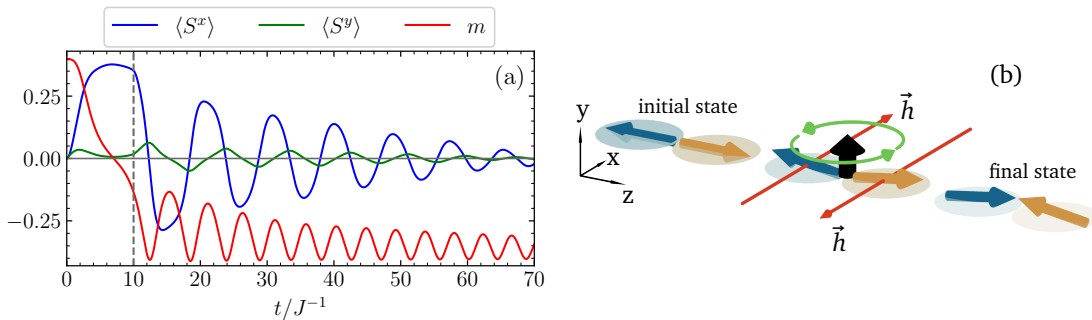

Figure 2: (a) The temporal evolution of spin expectation values under the effect of dephasing without relaxation. The switching field is present in the time interval $0 < t < 10J^{-1}$, as shown by the vertical gray dashed line marking the time until the field is applied, and its strength is $h = 0.08J$. The anisotropy parameter is $\chi = 0.9$ and temperature is set to zero. (b) The illustration of exchange-enhanced switching from $t = 0$ till switching. The initial state is shown by the arrows in the first circles on the left, the final state in the last circles on the right. The orange and blue arrows in the circles show the directions of the antiferromagnetic sublattice magnetizations. Applied staggered magnetic fields are shown with red arrows. The spins cant slightly after the magnetic field is applied, and at the same time they form a resulting strong effective field (black arrow) due to the exchange interaction. Consequently, the spins rotate (green curved arrows show the direction of rotation ) around the resulting effective field, i.e, the switching occurs.

## 4 Switching in dissipative quantum systems

No physical system is truly closed. Especially, since we are claiming the practical application of quantum antiferromagnets for data storage, the effect of an environment is inevitable. The energy transfer between spins and the lattice is of paramount importance in the control of sublattice magnetizations. For example, the interaction of localized spins with phonons in a lattice can affect the temporal evolution of the magnetization. Consequently, we analyze the dynamics of the sublattice magnetization taking into account the effect of the environment. The state of the open system changes as a consequence of its internal dynamics and of its interaction with the environment. Although we are not able to follow the dynamics of the environment, our goal is to understand its additional impact on the system of interest.

In this regard, we investigate the dynamics of the system that is weakly coupled to the bath using the Lindblad formalism, namely the adjoint quantum master equation [20] of the form

$$\frac{d}{dt}\langle O(t)\rangle = i\langle[\mathcal{H}, O(t)]\rangle + \sum_l \eta_l \left\langle \left( L_l^\dagger O(t) L_l - \frac{1}{2} O(t) L_l^\dagger L_l - \frac{1}{2} L_l^\dagger L_l O(t) \right) \right\rangle, \qquad (15)$$

for the expectation value of an observable $\langle O(t)\rangle$, where the Hamiltonian $\mathcal{H}$ corresponds to the system without environment. The Lindblad operators $\{L_l\}$ describe the system-bath interaction, and the parameters $\eta_l$ denote the relaxation rates and have the dimension of an inverse time. We choose the Lindblad operators such that $L_l$ excites the system by an energy $\omega_l$ and $L_l^\dagger$ de-excites it by the same energy. Then, to ensure convergence to the thermal equilibrium, the relaxation rates of excitation and de-excitation are related by the bosonic occupation number [25] and master equation reads

$$\begin{aligned}
\frac{d}{dt}\langle O(t)\rangle = {}& i\langle[\mathcal{H}, O(t)]\rangle + \frac{1}{2}\sum_l \eta_l \left( \langle[L_l, O(t)]L_l^\dagger\rangle + \langle L_l^\dagger[O(t), L_l]\rangle \right) \\
& + \frac{1}{2}\sum_l \eta_l n(\omega_l)\left( \langle[L_l, [O(t), L_l^\dagger]]\rangle + \langle[L_l^\dagger, [O(t), L_l]]\rangle \right),
\end{aligned} \qquad (16)$$

where $n(\omega_l)$ is the bosonic occupation function. Since we are using a bosonic representation to study the spin system, it is plausible that Lindblad operators modify the energy of the system for instance by creating and annihilating the Schwinger bosons, i.e., we treat the Schwinger bosons as the energy quanta of damped harmonic oscillators with $L_l = \alpha_{\mathbf{k}}^\dagger$ or $L_l = \beta_{\mathbf{k}}^\dagger$. This is illustrated in Figure 3. For simplicity, the damping rate is considered to be the same for all bosonic species, allowing only one damping parameter $\eta$.

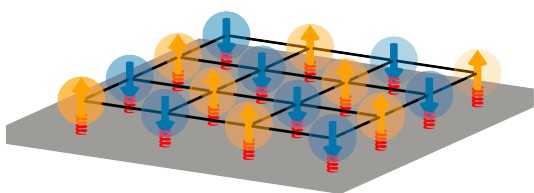

Figure 3: Illustration of an open quantum antiferromagnet. The red springs, connected to magnetization arrows, represent coupling of the spins to the environment, e.g. lattice vibrations and the gray layer represents the bath.

## 4.1 The equations of motions for the dissipative system

Having established the model, we can now compute a closed set of differential equations for the expectation values from Eq. (16) as follows

$$
\begin{cases}
\partial_t \langle a_{\mathbf{k}}^\dagger a_{\mathbf{k}} \rangle = i \langle [\mathcal{H}_{\mathrm{MF}}, a_{\mathbf{k}}^\dagger a_{\mathbf{k}}] \rangle + \eta \left( \frac{\lambda}{\omega_{\mathbf{k}}^-} \left( n(\omega_{\mathbf{k}}^-) + \frac{1}{2} \right) - \frac{1}{2} - \langle a_{\mathbf{k}}^\dagger a_{\mathbf{k}} \rangle \right), \\[2mm]
\partial_t \langle b_{\mathbf{k}}^\dagger b_{\mathbf{k}} \rangle = i \langle [\mathcal{H}_{\mathrm{MF}}, b_{\mathbf{k}}^\dagger b_{\mathbf{k}}] \rangle + \eta \left( \frac{\lambda}{\omega_{\mathbf{k}}^+} \left( n(\omega_{\mathbf{k}}^+) + \frac{1}{2} \right) - \frac{1}{2} - \langle b_{\mathbf{k}}^\dagger b_{\mathbf{k}} \rangle \right), \\[2mm]
\partial_t \langle a_{\mathbf{k}} a_{-\mathbf{k}} \rangle = i \langle [\mathcal{H}_{\mathrm{MF}}, a_{\mathbf{k}} a_{-\mathbf{k}}] \rangle + \eta \left( \frac{C_- \gamma_{\mathbf{k}}}{\omega_{\mathbf{k}}^-} \left( n(\omega_{\mathbf{k}}^-) + \frac{1}{2} \right) - \langle a_{\mathbf{k}} a_{-\mathbf{k}} \rangle \right), \\[2mm]
\partial_t \langle b_{\mathbf{k}} b_{-\mathbf{k}} \rangle = i \langle [\mathcal{H}_{\mathrm{MF}}, b_{\mathbf{k}} b_{-\mathbf{k}}] \rangle + \eta \left( \frac{C_+ \gamma_{\mathbf{k}}}{\omega_{\mathbf{k}}^+} \left( n(\omega_{\mathbf{k}}^+) + \frac{1}{2} \right) - \langle b_{\mathbf{k}} b_{-\mathbf{k}} \rangle \right), \\[2mm]
\partial_t \langle a_{\mathbf{k}}^\dagger b_{\mathbf{k}} \rangle = i \langle [\mathcal{H}_{\mathrm{MF}}, a_{\mathbf{k}}^\dagger b_{\mathbf{k}}] \rangle - \eta \langle a_{\mathbf{k}}^\dagger b_{\mathbf{k}} \rangle, \\[2mm]
\partial_t \langle a_{\mathbf{k}} b_{-\mathbf{k}} \rangle = i \langle [\mathcal{H}_{\mathrm{MF}}, a_{\mathbf{k}} b_{-\mathbf{k}}] \rangle - \eta \langle a_{\mathbf{k}} b_{-\mathbf{k}} \rangle.
\end{cases}
\tag{17}
$$

The first commutators in all the above equations have already been obtained in Eqs. (13). This solvable closed set of differential equations enables us to analyze the magnetization switching behavior of the quantum antiferromagnet coupled to the environment.

However, we treat the system as a damped harmonic oscillator [20], which results in a decrease of the bosonic occupation number, which no longer satisfies the constraint in Eq. (4). Moreover, bosonic operators are time-dependent in our model, and they modify the bosonic occupation number in a time-dependent manner. Therefore, we incorporate a time-dependent Lagrange parameter $\lambda$ in our approach to compensate for these fluctuations. One can adjust $\lambda$ so that the total number of bosons remains constant, and the constraint in the bosonic occupation number is fulfilled on average in each sublattice as

$$
\frac{1}{N} \sum_{\mathbf{k}} \left( \langle a_{\mathbf{k}}^\dagger a_{\mathbf{k}} \rangle + \langle b_{\mathbf{k}}^\dagger b_{\mathbf{k}} \rangle \right) = 2S.
\tag{18}
$$

In this framework, we construct another differential equation by summing the first two equations in the Eqs. (17) over all momenta $\mathbf{k}$

$$
0 \stackrel{!}{=} \partial_t \left( \frac{1}{N} \sum_{\mathbf{k}} \left( \langle a_{\mathbf{k}}^\dagger a_{\mathbf{k}} \rangle + \langle b_{\mathbf{k}}^\dagger b_{\mathbf{k}} \rangle \right) \right) = i \left\langle \left[ \mathcal{H}_{\mathrm{MF}}, \frac{1}{N} \sum_{\mathbf{k}} \left( a_{\mathbf{k}}^\dagger a_{\mathbf{k}} + b_{\mathbf{k}}^\dagger b_{\mathbf{k}} \right) \right] \right\rangle
$$
$$
+ \frac{\eta}{N} \sum_{\mathbf{k}} \left( \frac{\lambda}{\omega_{\mathbf{k}}^-} \left( n(\omega_{\mathbf{k}}^-) + \frac{1}{2} \right) - \frac{1}{2} + \frac{\lambda}{\omega_{\mathbf{k}}^+} \left( n(\omega_{\mathbf{k}}^+) + \frac{1}{2} \right) - \frac{1}{2} \right) - \frac{\eta}{N} \sum_{\mathbf{k}} \left( \langle a_{\mathbf{k}}^\dagger a_{\mathbf{k}} \rangle + \langle b_{\mathbf{k}}^\dagger b_{\mathbf{k}} \rangle \right).
\tag{19}
$$

The first term on the right hand side vanishes and the last term in the second line is constant. By differentiating the remaining terms on the second line with respect to time, we obtain the following equation

$$
\frac{\mathrm{d}}{\mathrm{d}t} \underbrace{\left[ \frac{\eta}{N} \sum_{\mathbf{k}} \left( \frac{\lambda}{\omega_{\mathbf{k}}^-} \left( n(\omega_{\mathbf{k}}^-) + \frac{1}{2} \right) - \frac{1}{2} + \frac{\lambda}{\omega_{\mathbf{k}}^+} \left( n(\omega_{\mathbf{k}}^+) + \frac{1}{2} \right) - \frac{1}{2} \right) \right]}_{D} = \frac{\mathrm{d}D}{\mathrm{d}t} = 0,
\tag{20}
$$

where we have again used that the total occupation number does not change in time according to the constraint in Eq.(18). At this stage, it is important to note that the mean-field averages

$A$ and $B$ in Eq.(9) are also time dependent. Therefore, one can see that $D = D(A, B, \lambda)$ which leads to the following new differential equation

$$\frac{\mathrm{d}\lambda}{\mathrm{d}t} = -\frac{1}{\frac{\partial D}{\partial \lambda}} \left( \frac{\partial D}{\partial A} \frac{\mathrm{d}A}{\mathrm{d}t} + \frac{\partial D}{\partial B} \frac{\mathrm{d}B}{\mathrm{d}t} \right). \tag{21}$$

This completes the required set of differential equations that can be solved for each momentum with the time-dependent Lagrange parameter. Furthermore, in order to determine a suitable initial value of the magnetization that is compatible with the infinite system,[3] we treat the initial system such that there are more "$a$" Schwinger bosons than "$b$" type. This is reached by condensation of one boson flavor [22]. For the finite-size system at zero temperature, this corresponds to a very tiny energy gap for the former boson flavor and a large energy gap for the latter ($\Delta^+ \gg \Delta^- \geq 0$).

In the solution, numerical instabilities might occur in the equations because the expression for the energy gaps has the form

$$\Delta^\pm = \sqrt{\lambda^2 - \left| C_\pm \right|^2}. \tag{22}$$

The gaps $\Delta^\pm$ appear in denominators of expectation values in Eq. (12) at $\mathbf{k} = \mathbf{k}_0 = (0, 0)$ and $\mathbf{k}_0 = (\pi, \pi)$. These modes contribute macroscopically to the occupation of bosons, i.e., they scale with the size of the cluster which is an unexpected feature of single modes. Thus, it is essential to carefully consider this aspect while solving differential equations. Nevertheless, we have now established all the analytical relations necessary to solve the equations at each $\mathbf{k}$ point in the two-dimensional Brillouin zone and thereby determine the non-equilibrium properties of the system. There one still needs to deal with two-dimensional sums, which are integrals in the thermodynamic limit. However, all equations for the expectation values in (11), (12) and differential equations in (17) depend on $\gamma_\mathbf{k}$, not explicitly on $\mathbf{k}$. Hence, we replace the integrals over all wave vectors with a single integral over $\gamma$ and discretize the $\gamma \in [-1, 1]$ space for the efficiency of the numerical calculations, provided that the density of states is known. Details of the transformation can be found in Refs. [17, 18]. Note that our numerical calculations are consistent with results obtained through summation over the first Brillouin zone of the square lattice for a system size of 500 spins. For systems of this size, assuming spontaneous symmetry breaking neglecting quantum tunneling between the two Néel-type orderings is justified. This is realistic number even in nano-scale samples. To compare, assume $10\,\mathrm{nm} \times 10\,\mathrm{nm}$ size antiferromagnet in 2d, for example antiferromagnet NiO with the lattic constant a=4.176 Å [26]. Then, this small 2d NiO antiferromagnet contains approximately 575 sites. Therefore, quantum many-body theory is required so simulate that large system, and quantum tunneling can be disregarded though it matters for systems with $\mathcal{O}(20)$ spins [27]. Furthermore, the fourth-order Runge-Kutta algorithm is employed to obtain the numerical solution of the differential equations with the time step $\Delta t = 10^{-4} J^{-1}$. This small step size is chosen to mitigate the potential numerical instabilities arising from the presence of the tiny gaps of the specific $\mathbf{k}_0 = (0, 0)$ and $(\pi, \pi)$ modes.

## 5 Results and discussions

### 5.1 The sublattice magnetization dynamics with relaxation at zero temperature

Now, we start to analyze the non-equilibrium properties of antiferromagnets coupled to a bath, as a result of Eqs. (17) and (21). A finite, non-zero staggered field is applied in the interval of

---

[3]The desired initial sublattice magnetization for a square lattice at zero temperature is $m_0(\chi = 0) = 0.3034$ [22].

$0 < t < 10\,J^{-1}$ in all analyses. Indeed, this is the case in practical applications as one aims at switching with finite pulses. For typical antiferromagnetic exchange couplings, this time scale still corresponds to the THz regime, meaning $t \leq 1\,\text{ps}$.

Figure 4 sketches one of our main results in this work. The dynamics of the bosonic occupation numbers and the sublattice magnetization are shown in Figure 4(a). According to Eq.(5), the mean occupation numbers of the Schwinger bosons define the spin expectation value $\langle S^z \rangle$ and thus also the sublattice magnetization $m$. So we expect the Schwinger bosons' occupations to be swapped during the switching. Since condensation of the $a$ bosons is assumed in the initial state (orange line), the $b$ bosons should be condensed (blue line) after switching the sublattice magnetization. Most importantly, the relaxation results in almost static sublattice magnetization immediately after full switching (red line). The constraint in Eq. (4) is also fulfilled at all times (dashed green line). A similar plot to that in Figure 2(a) is presented in Figure 4(b), but now with relaxation. One can see the immediate convergence of the order to the reoriented stable magnetization state with a remaining very weak decaying Larmor precession. This is precisely how the dissipation is expected to impact the spin dynamics. Furthermore, the time evolution of the spin expectation values supports the realization of exchange enhancement in the system as it was illustrated in Figure 2(b). Our mean-field approach is therefore able to accurately model exchange enhancement in switching processes, including dephasing and relaxation.

Figure 5(a) shows the sublattice magnetization dynamics at $\chi = 0.9$ for different relaxation rates. One can see that in all cases the oscillations decay after switching and the damping rate is essentially given by the decay rate. The system without damping ($\eta = 0$), i.e., the effects of dephasing only, displays a very slow decay while maintaining continuous magnetization oscillations (red line in Figure 5(a)). On the other hand, spin-lattice relaxation derived from the Lindblad formalism induces the relaxation of the oscillation exponentially. Eventually, the system with stronger dissipation yields fast convergence to the static state with full switching (zoom in the plot in Figure 5(a)). Moreover, the order quickly reaches a coherent static state that is opposite to the initial order, for example at $\eta = 0.08\,J$. Note that dissipation increases the threshold value of the necessary switching field compared to the case of $\eta = 0$. This is natural, as dissipation decrements the energy of the system and thus affects the non-equilibrium dynamics. Therefore, we choose the switching field so that the switching occurs for all the selected relaxation rates. For example, the threshold value of the staggered field is actually $h_{\text{thr}} = 0.079\,J$ at the anisotropy of $\chi = 0.9$ for a closed antiferromagnetic square lattice [18], but we have chosen $h = 0.094\,J$ in Figure 5(a).

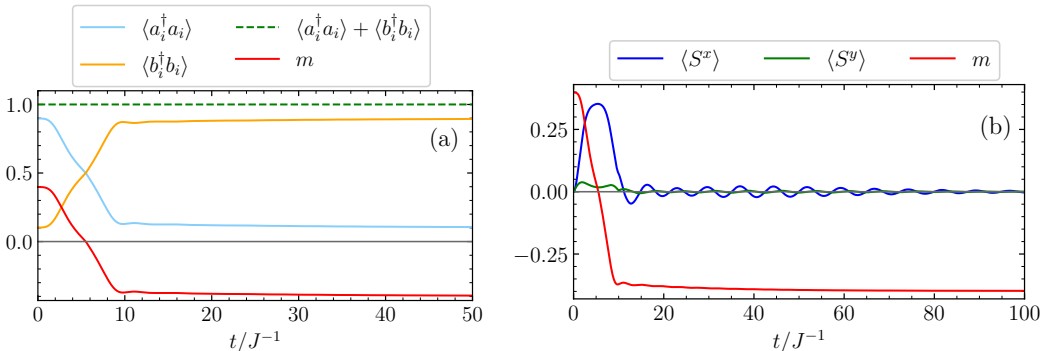

Figure 4: (a) The dynamics of the mean occupation of the Schwinger bosons, as well as the resulting sublattice magnetization. (b) The time evolution of the spin expectation values under the effect of relaxation for the rate $\eta = 0.05\,J$. The applied external field is $h = 0.09\,J$ and the anisotropy parameter is $\chi = 0.9$.

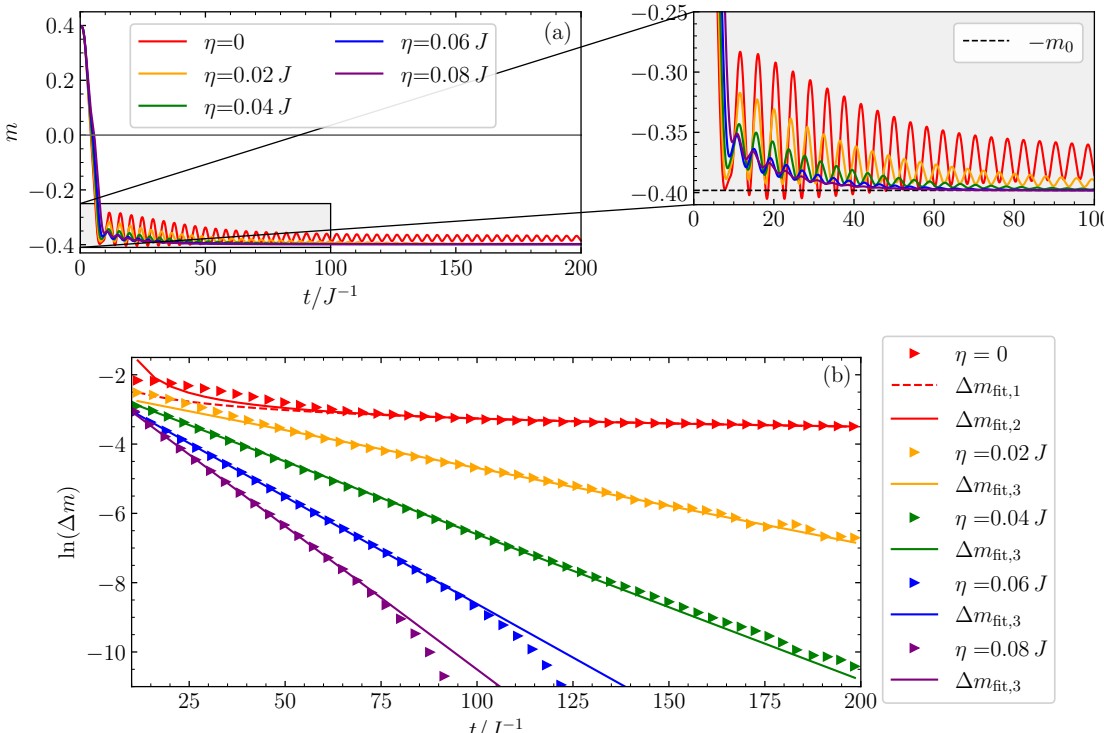

Figure 5: (a) The dynamics of the sublattice magnetization of an antiferromagnet coupled to the environment for different relaxation rates at $\chi = 0.9$. The static staggered magnetic field is present in the time interval $0 < t < 10 J^{-1}$ with the value $h = 0.094 J$. The zoom is included to show the decay of oscillations more clearly at different decay rates. The panel (b) shows the damping of the oscillations of the magnetization in (a). We define $\Delta m = m_0 - |m_{\max}(t_i)|$ where $m_0$ is the initial magnetization at $t = 0$ and $m_{\max}(t_i)$ is the value of the magnetization when the oscillations reach a peak at time $t_i$. The fitting functions are: $\Delta m_{\text{fit},1} = C/t^{\alpha_{\text{fit},1}}$ with $\alpha_{\text{fit},1} \approx 0.345$ and $\Delta m_{\text{fit},2} = C/\ln(\alpha_{\text{fit},2} t)$ with $\alpha_{\text{fit},2} \approx 0.142 J$ ; $\Delta m_{\text{fit},3} = C e^{-\eta_{\text{fit}} t}$ with $\eta_{\text{fit}} \approx 0.022 J$ for $\eta = 0.02$ data, $\eta_{\text{fit}} \approx 0.042 J$ for $\eta = 0.04$, $\eta_{\text{fit}} \approx 0.061 J$ for $\eta = 0.06$ and $\eta_{\text{fit}} \approx 0.082 J$ for $\eta = 0.08$ case. The fitting for $\eta = 0$ is done in the interval $75 J^{-1} < t < 200 J^{-1}$. The nonlinear part of the data in $\eta = 0.06 J$ and $\eta = 0.08 J$ are neglected during the fitting, because very small values appear due to the numerical inaccuracies only.

For the sake of completeness, we provide Figure 5(b) to validate the qualitative difference in the damping between the closed system and the system coupled to the bath. The change in amplitude of magnetization, denoted by $\Delta m = m_0 - |m_{\max}(t_i)|$, is extracted from Figure 5(a), where $m_{\max}(t_i)$ is the value of magnetization at the maxima of oscillations at time $t_i$ and $m_0$ is the initial magnetization. Note that Figure 5(b) has a logarithmic scale on the $y$ axis. An alternative plot is presented in the Appendix A with both axes in logarithmic scale. On the one hand, one can see that the closed system shows a very slow decrease in $\Delta m$ (the data with red triangles in Figure 5(b)). The fitting with the power law function $\Delta m_{\text{fit},1} = C/t^{\alpha_{\text{fit},1}}$ also confirms its steady decrease with a small exponent value of $\alpha_{\text{fit},1} \approx 0.345$ (solid red line). Furthermore, another possible fit function $\Delta m_{\text{fit},2} = C/\ln(\alpha_{\text{fit},2} t)$ works equally well and supports our claim of a very slow, non-exponential damping solely due to dephasing with $\alpha_{\text{fit},2} \approx 0.142 J$ (dashed red line). On the other hand, the system with finite relaxation shows an exponential decrease in $\Delta m$. Indeed, the curves are fitted by the exponential function $\Delta m_{\text{fit},3} = C e^{-\eta_{\text{fit}} t}$,

and the fitting parameter $\eta_{\text{fit}}$ is in close agreement with the corresponding decay rate in the Lindblad formalism. For example, the data for the coupling of $\eta = 0.04J$ shows an exponential decay of oscillations in magnetization with a fitting parameter of $\eta_{\text{fit}} \approx 0.042J$ in the exponent (green triangles and green line in Figure 5(b)). Similar analyses for the anisotropy parameter of $\chi = 0.98$ are provided in Figure 8 in Appendix B, thereby substantiating the conclusions of this subsection. Consequently, these findings ensure the reliability of our quantum approach in capturing dephasing and relaxation on equal footing.

In fact, the observed exponential decay in oscillations of switched magnetization is actually a combination of dephasing and relaxation effects. Indeed, we couple the quantum antiferromagnet to the environment, which already shows the effects of dephasing. Therefore, the relaxation of sublattice magnetization toward the next energetically favorable orientation is the result of internal and external quantum effects, i.e., dephasing and spin-lattice relaxation. These are really promising observations of our quantum theoretical model because the static, non-oscillatory state after switching in antiferromagnets guarantees ultrafast and safe data storage in practice. We claim that there is a strong potential for experimental realizations.

## 5.2 The singular behavior in relaxation

The switching process is characterized by closing the spin gap in Eq. (10) at the instant of switching, followed by its reopening afterwards as the Néel vector reorients. Therefore, for completeness, we also analyze the dynamics of the energy gaps in Eq. (22).

The dominating contribution comes from the $\mathbf{k}_0 = (0,0)$ and $(\pi,\pi)$ modes in the differential equations in (17) because we started our simulations from the state with macroscopic occupation of one boson type ($a$ bosons in our initialization). This implies that there is only a tiny energy gap at the points $\mathbf{k}_0$. The minimum excitation energy reduces to a small value

$$\lim_{\mathbf{k}\to\mathbf{k}_0} \omega^-(\mathbf{k}) = \Delta^- \approx \frac{J}{M}, \tag{23}$$

where $M$ is the number of points in discretized $\gamma$ space and we consider up to $M = 500$ points in our simulations. These gaps appear in the denominators in dissipation part of the differential equations in (17). Therefore, singularities can emerge while solving them together with Eq. (21). We observe numerical instabilities, that is, anomalously large values scaling with the system size, in the solutions at specific high-symmetry points, particularly at $\mathbf{k}_0 = (0,0)$ and $(\pi,\pi)$. Indeed, these points correspond to the zone center and the zone boundary, respectively, and are associated with singular behavior in our model due to dissipation. In plain words, the macroscopically occupied modes are damped particularly strongly. Already an arbitrary small amount of dissipation leads to a qualitatively different behavior. As a result of this singularity, the macroscopic occupation of one bosonic flavor at these modes is always present in the sublattice at all times. This is a characteristic feature of dissipative switching in our model. Figure 6 confirms this behavior with the dynamics of energy gaps $\Delta^-$ and $\Delta^+$ for $a$ and $b$ Schwinger bosons, respectively. In the system without dissipation, the energy gap for the $a$ bosons continuously increases, whereas it starts to decrease continuously for the $b$ bosons and they intersect in the course of switching (dashed lines). The interchange of bosons occurs after switching. In contrast, the dynamics of the energy gaps are qualitatively different for the dissipative system. Initially, the macroscopic occupation of $a$ bosons is present in the system with a very small gap of $\Delta^-(t=0) \approx 2 \cdot 10^{-3} J$ and this tiny gap remains almost unchanged (solid green line) until magnetization reaches zero. Then, the macroscopic occupation occurs for the $b$ bosons after switching (solid red line). Another significant difference is that both gaps asymptotically approach distinct, quasi-stationary values in the dissipative case, and no further oscillations occur. Again, this phenomenon can be attributed to the effect of dissipation induced by the environment. Actually, Figure 4 and the solid lines in Figure 6 describe the same

physical process in different quantities. The dynamics of the energy gaps and the change in the bosonic occupation numbers are totally consistent in these graphs, in accordance with the crossings of different bosonic occupations in the course of switching.

The effective energy gaps increase at finite temperature due to thermal fluctuations [16]. However, the distinct behavior of the gaps with or without dissipation persists there as well. We provide finite-temperature results for the energy gap in the Appendix C, including magnetization dynamics at different system-bath coupling strengths.

So far, we have used the simplest case of relaxation by assuming Schwinger bosons as damped harmonic oscillators. Furthermore, more complex dissipation can be considered as well to analyze the existence of singular effects in more depth. For instance, the decay rate can acquire a momentum dependence or the relaxation can be chosen to conserve the total spin. But this is beyond the scope of the present work and left to future research.

# 6 Conclusion

We studied the switching processes in quantum antiferromagnetic square lattice with spin-1/2, coupled to an environment. The approach is based on time-dependent Schwinger boson mean-field theory, and the effect of the environment is derived from the Lindblad formalism. The primary objective of the approach was to obtain efficient and stable reorientation of the antiferromagnetic order and to highlight the distinct effects of dephasing and relaxation in switching processes.

Our simulations, which incorporate all magnonic modes in a large finite system with all wave vectors, enable us to capture the effects of dephasing which manifests itself as a slow power-law decrease in the magnetization oscillations after switching. In addition, the effect of the dissipation on the dynamics of the magnetization was found to be of paramount importance because spin-lattice relaxation induces a fast exponential decay of the oscillations. The reoriented magnetization dynamics settle into a static state, which indicates that post-switching oscillations in the sublattice magnetization are completely suppressed due to the dissipation after a short time. However, a singular behavior was observed in the switching process with relaxation, characterized by the significantly strong damping of particular modes with their macroscopic occupation. This phenomenon requires further investigation, partic-

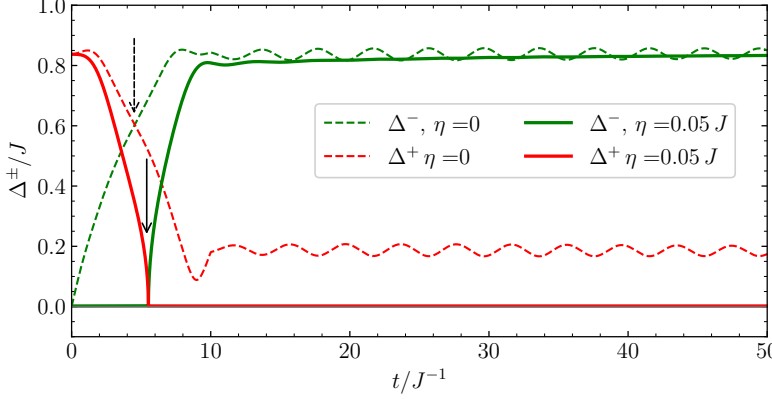

Figure 6: The time evolution of the energy gaps with and without relaxation for $\chi = 0.9$. The applied switching field strength is $h = 0.09\,J$. The solid and dashed vertical black arrows indicate the instant of switching with and without relaxation, respectively.

ularly in the context of more complex dissipation, given our present treatment of Schwinger bosons as energy quanta of damped harmonic oscillators.

Our results show that exchange-enhancement allows for low external fields to switch the sublattice magnetization and relaxation can drive coherent magnon dynamics in open quantum antiferromagnets, thereby enabling ultrafast reorientation of the magnetization without post-switching oscillations. Indeed, this is the central goal of antiferromagnetic spintronics to store the information efficiently and safely at the THz regime. The stable switched state is obtained under the field value of $h = 0.09J$ for the anisotropy of $\chi = 0.9$ and the relaxation rate of $\eta = 0.05J$, which corresponds to roughly 8 T if we assume $J = 10$ meV. The time taken for the full switching $t \approx 10J^{-1}$ corresponds to $\approx 0.65$ ps in our results, which is within THz regime. One can still reduce the threshold field by applying time-dependent control fields at resonance frequency [17] and at weaker anisotropies [18]. Nevertheless, the estimated numbers already indicate the realistic possibility of the observation in the laboratory to confirm efficient control of sublattice magnetization at ultrafast scale. We do not think that the Landau-Zener-Stückelberg effect well-known for two-level system [28] is a promising alternative with respect to switching speed and completeness. In summary, these results and the method developed to obtain them pave the way to a better understanding of magnetization dynamics and hence to sustainable information processing based on quantum antiferromagnetism.

These findings provide key insights for the development of novel spintronic devices. The fast and robust switching in this study is very promising result for practical applications. The topic requires deeper theoretical investigations to further explore the magnetization control for specific systems. One can consider different anisotropies of the quantum antiferromagnets with higher spins. The combination of external alternating and uniform fields is another possible issue to tackle. In addition, alternative numerical approaches are needed to support the findings of mean-field approximations that have been used mainly so far.

## Acknowledgments

We are thankful to T. Gräßer and P. Bieniek for useful discussions and suggestions.

**Funding information**   This work has been funded by the Deutsche Forschungsgemeinschaft (German Research Foundation) in project UH 90/14-2.

## A   Distinct effect of dephasing and relaxation on switching

To show the decay of oscillations with and without dissipation more clearly, we provide an alternative version of Figure 5(b). The same data are depicted in Figure 7 in a log-log plot. One can clearly see the difference between the effect of dephasing and relaxation. The dephasing results in a slow power-law decrease in the oscillations whereas the relaxation shows an exponential fast decline. The power law fitting $\Delta m_{\mathrm{fit},1} = C/t^{\alpha_{\mathrm{fit},1}}$ with the small decay exponent of $\alpha_{\mathrm{fit},1} \approx 0.345$ also confirms the slow decay of oscillations in sublattice magnetization of the antiferromagnet without dissipation.

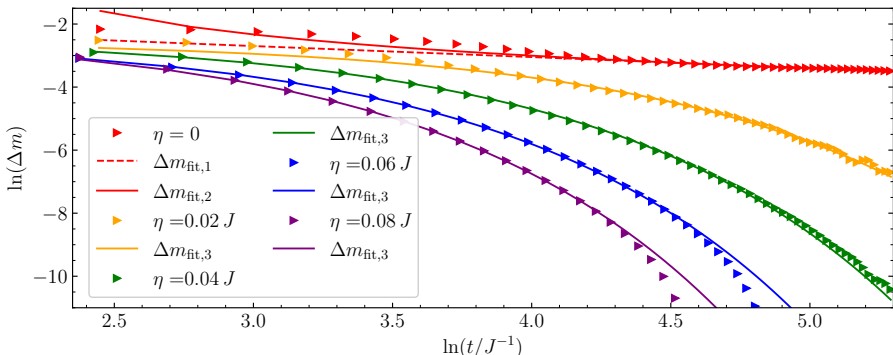

Figure 7: The decrease in sublattice magnetization oscillations with the same parameters as in Figure 5(b), but with logarithmic scale on both axes. Fitting functions: $\Delta m_{\text{fit},1} = C/t^{\alpha_{\text{fit},1}}$ with $\alpha_{\text{fit},1} \approx 0.345$ and $\Delta m_{\text{fit},2} = C/\ln(\alpha_{\text{fit},2}t)$ with $\alpha_{\text{fit},2} \approx 0.142J$; $\Delta m_{\text{fit},3} = Ce^{-\eta_{\text{fit}}t}$ with the same $\eta_{\text{fit}}$ values from Figure 5(b).

## B  Switching in dissipative system at low anisotropy

Here we present the results for a different anisotropy ($\chi = 0.98$) and concomitantly at lower switching field ($h = 0.025\,J$), but with the same value for the decay rate parameters as in Figure 5. In fact, other values of the anisotropy parameter have been studied as well, and it was found that they also corroborate the results presented in the main text. Therefore, there exists a large interval of anisotropy parameters to obtain and to confirm the main conclusions of the work.

The dynamics of the magnetization is slower at low anisotropies because of the required lower switching fields, as one can see from Figure 8(a). Furthermore, the $\eta = 0$ case clearly shows the effect of dephasing after switching with power-law decrease in the oscillations (red line in Figure 8(a) and red triangles in Figure 8(b)). The fitting function $\Delta m_{\text{fit},1} = C/t^{\alpha_{\text{fit},1}}$ also implies a small exponent, where $\alpha_{\text{fit},1} \approx 0.531$ for solid red line in Figure 8(b). The inclusion of relaxation decreases the oscillations more rapidly and eventually implies saturation in a steady-state. Moreover, the fitting function $\Delta m_{\text{fit},3}$ provides in close agreement between fitting parameters $\eta_{\text{fit}}$ and the actual dissipation rates $\eta$ (see the caption in Figure 8). Figure 8(c) presents clearly distinct effects of dephasing and relaxation with their power-law and exponential decrease on $\Delta m$, respectively.

## C  Finite temperature

The behavior of switching in open quantum antiferromagnet at finite temperature is highly relevant when it comes to practical applications. At finite temperature, thermal fluctuations contribute to the reduction of the spin gap, thereby a lower threshold magnetic field is required for magnetization switching [16].

Figure 9(a) demonstrates the dynamics of the sublattice magnetization at $T = 0.4\,J$ for the anisotropy parameter of $\chi = 0.9$, where $T_{\text{Neel}} = 0.704\,J$ holds at this particular value of the anisotropy [16]. Slightly lower fields are sufficient to obtain switching compared to the zero temperature results in Figure 5(a). It can be seen that the system that is not coupled to the environment shows small-amplitude oscillations around the reoriented static state. However, oscillations decrease considerably faster after switching as the decay rates increase. Moreover, the steady-state of the system after switching occurs almost immediately in an open quantum system due to spin-lattice relaxation.

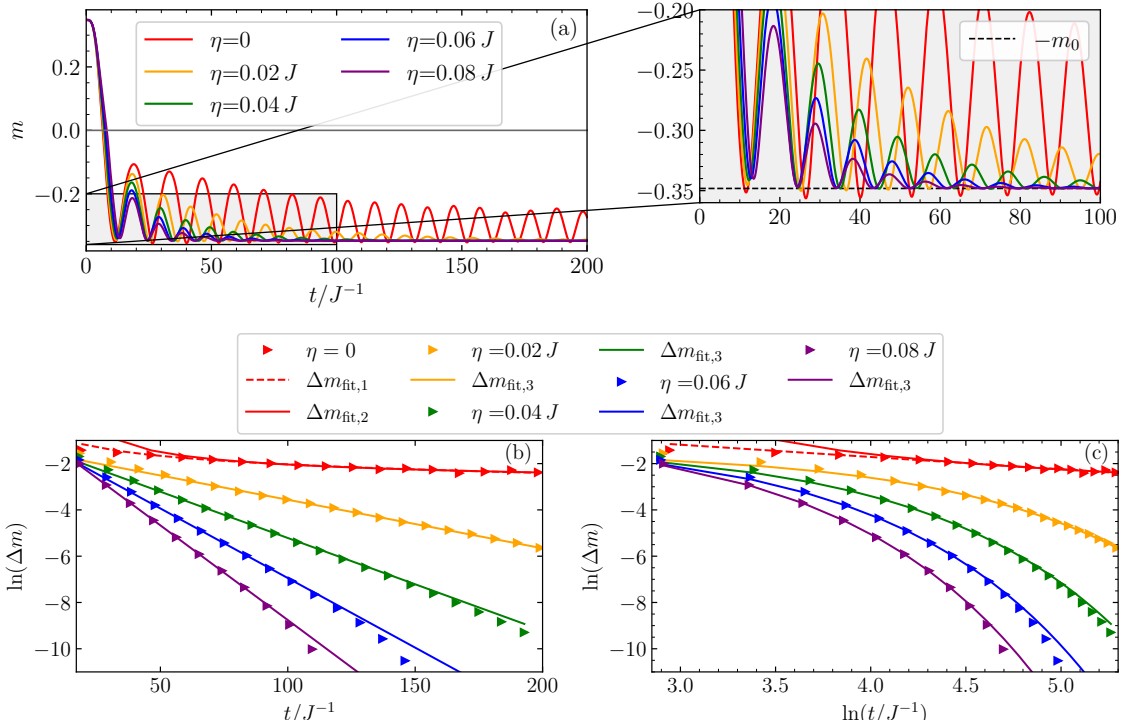

Figure 8: (a) The dynamics of the sublattice magnetization of an antiferromagnet coupled to the environment for different decay rates at $\chi = 0.98$. The static staggered magnetic filed is applied in the time interval $0 < t < 10 J^{-1}$ with the value $h = 0.025 J$. The zoom is included to show the decay of the oscillations more clearly at different rates. Panel (b) shows the damping of the oscillations of magnetization in (a). $\Delta m$ is defined in the same way as in Figure 5. The fitting functions read: $\Delta m_{\text{fit},1} = C/t^{\alpha_{\text{fit},1}}$ with $\alpha_{\text{fit},1} \approx 0.531$ and $\Delta m_{\text{fit},2} = C/\ln(\alpha_{\text{fit},2} t)$ with $\alpha_{\text{fit},2} \approx 0.054 J$ ; $\Delta m_{\text{fit},3} = C e^{-\eta_{\text{fit}} t}$ with $\eta_{\text{fit}} \approx 0.021 J$ for $\eta = 0.02$ data, $\eta_{\text{fit}} \approx 0.04 J$ for $\eta = 0.04$, $\eta_{\text{fit}} \approx 0.061 J$ for $\eta = 0.06$ and $\eta_{\text{fit}} \approx 0.082 J$ for $\eta = 0.08$ case. The fitting for $\eta = 0$ is done in the interval $75 J^{-1} < t < 200 J^{-1}$. The downturn of the data in $\eta = 0.06 J$ and $\eta = 0.08 J$ is neglected in the fitting, since it appears at very small values due to numerical issues. Panel (c) contains the same data as panel (b), but with logarithmic scale on both axes.

The singular behavior of the relaxation persists at finite temperature as well, but with larger spin gaps. Figure 9(b) illustrates the temporal dynamics of the energy gaps $\Delta^-$ and $\Delta^+$ under two distinct conditions: a closed system ($\eta = 0$) and an open system coupled to an environment ($\eta = 0.08 J$). The energy gap values are $\Delta^-(t = 0) \approx 0.0243 J$ and $\Delta^+(t = 0) \approx 0.8 J$ and are essentially interchanged upon switching. Initially, the two gaps evolve in opposite directions: one gap decreases while the other increases. In contrast, the macroscopic occupation of one bosonic mode at corresponding momenta $\mathbf{k}_0 = (0,0)$ and $(\pi, \pi)$ is present in the dissipative system over the entire time evolution while these occupations evolve gradually in the absence of relaxation.

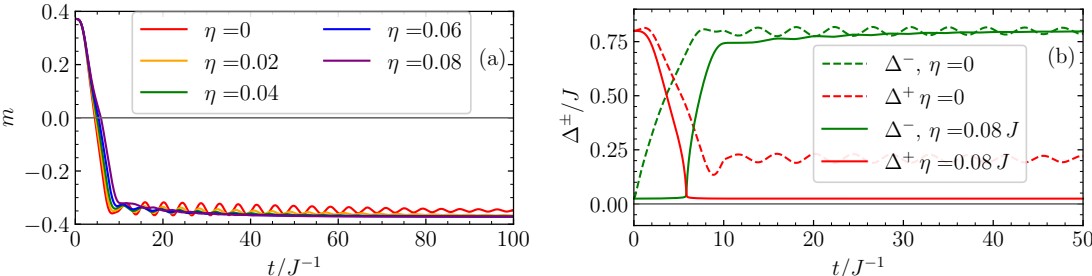

Figure 9: (a) The dynamics of the magnetization for different damping parameters at $T = 0.4J$, and $\chi = 0.9$. The switching field is static with the strength of and $h = 0.09J$ and it is present in the interval $0 < t < 10J^{-1}$. (b) The temporal evolution of the spin gap for the closed (dashed lines) and open (solid lines) quantum antiferromagnetic square lattice with the same set of parameters as in panel (a).

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
