# Peer review of "The effect of dephasing and spin-lattice relaxation during the switching processes in quantum antiferromagnets"

_SciPost Physics, doi:SciPost Phys. 19, 117 (2025)_

## Round 1 · Referee Report · Anonymous (Referee 1) · 2025-9-8

Strengths

This is a very interesting paper on magnetic storage technology that rests on driving antiferromagnets. It is clearly written and well approachable by a general audience. The theoretical approach with its approximation seems justified.

Weaknesses

I have some recommendations regarding the introduction, a few minor points and some additional literature.

Report

In my opinion the manuscript meets the acceptance criteria of the journal. I would prefer a few additional remarks for the broader audience in the introduction. The I have some technical improvements and would like to suggest reference to some literature.

Requested changes

  1. In the introduction I would welcome a short explanation how the bit is read out in an antiferromagnetic realization. Everybody understands how a magnetization can be read out, but how is the encoded bit read out for the antiferromagnet?

  2. How would the system behave under miniaturization? The Neel vector is not an eigenstate of the quantum Hamiltonian and would thus tunnel, see e.g., Waldmann et al., Phys. Rev. Lett. 102, 157202. Could you please comment on this problem?

  3. I would also welcome if you could comment on non-linear or non-constant drivings of the field as suggested by Garanin and Schilling, see e.g., Garanin, D. A. and Schilling, R., Phys. Rev. B 66, 174438.

  4. Line 40, word missing in front of [40]

  5. Caption of Fig. 1, typo in first line: tan -> an

  6. Eq. (2) and line 112; please explain single index in 2-d system. Index i in equations always potentially misleading.

  7. Lines 105 & 107: in -> of?

  8. Lines 120 & 121: E_0 is the constant energy or a constant energy?

  9. Figure 2: I suggest a vertical dashed line to mark the time until the field is applied.

  10. Lines 329 & 337: I find \vec{k}_0=0,\vec{\pi} unclear. Please explain.

  11. The caption of Fig. 6 speaks of "solid and dashed vertical black arrows", but I do not see any.

  12. The article is written in clear English. If the authors want they could insert a few "the" and commas for main clauses beginning with and.

Recommendation

Ask for minor revision

  • validity: high
  • significance: high
  • originality: high
  • clarity: high
  • formatting: excellent
  • grammar: excellent

Author:  Asliddin Khudoyberdiev  on 2025-09-16  [id 5822]

(in reply to Report 1 on 2025-09-08)
Category:
remark
answer to question
correction
pointer to related literature

Thank you for your kind feedback. We are thrilled to hear that you found the paper as “clearly written and accessible to the general reader”. Indeed, making the topic approachable was an important goal. We appreciate your remarks on the theoretical approach and useful suggestions including beneficial literature. We believe that the changes have increased the readability of our manuscript. Action taken: All the changes are highlighted in red.

Addressing requested changes:

  1. We fully agree with this suggestion and included a brief explanation in the first paragraph of the introduction on how the bit is read out in an antiferromagnetic realization. As you pointed out, while the readout mechanism for ferromagnets is widely understood, it is important to clarify how information can be retrieved in the antiferromagnetic case to ensure clarity for all readers. Action taken: We included additional explanations and references for read out techniques in antiferromagnets in the second paragraph of the introduction.

  2. We appreciate the comment and provided further clarification. First, let us stress that we do not consider the Néel state to be the ground state. The mean-field approach does capture quantum fluctuations around the Néel ordering by Bogoliubov transformations. We agree, however, that quantum tunneling between the two possible Néel orderings is not captured by our approach. Yet, such tunneling is relevant only for truly microscopic systems with O(20) spins. However, we consider large systems where we discretize the density of gamma space considering 500 gamma points corresponding roughly to systems with 500 spins and more within the numerical accuracy. For comparison, let's assume 10nm x 10nm lattice in 2d, for example NiO antiferromagnet with the lattice constant a=4.17 Å=0.417 nm. Hence, the area per 2d unit cell equals ≈ 0.174 nm2 and the area of the 2d lattice=100 nm2 so that the number of spins results to be (Area of 2d lattice)/(area per 2d unit cell)≈575. Hence, even nano-scale samples contain O(500) spins for which spontaneous symmetry breaking is a valid concept. Action taken: We provided numerical estimates for small 2d antiferromagnet and emphasized the possibility of Neel vector tunneling only in small systems at the end of the fourth section by citing Waldmann et al., Phys. Rev. Lett. 102, 157202 (2009).

  3. Our framework allows for the implementation of nonlinear or non-constant field drivings, similar in spirit to those analyzed by Garanin and Schilling [Phys. Rev. B 66, 174438 (2002)] in the context of the Landau-Zener-Stückelberg effect. Indeed, in our previous publication we considered time-dependent pulse at resonance frequency [15] and obtained the switching at lower fields. We believe that this opens a path toward optimizing switching protocols in experiment based on our theoretical predictions. But we do not think that the Landau-Zener-Stückelberg effect is an efficient way to obtain switching which should be carried out as fast and as complete as possible. Action taken: In the third paragraph of the conclusion, we mentioned that “One can still reduce the threshold field by applying time-dependent control fields at resonance frequency [17]”. Additionally, we commented on the Landau-Zener-Stückelberg effect by citing Ref. [28].

  4. We are sorry for the missing word there. Actually, the switching occurs under low fields in the case of exchange enhancement. This is what we want to highlight there. Action taken: We corrected the sentence in front of Ref. [18].

  5. We are very thankful for your careful reading. Indeed, this is a typo and it is actually an article. Action taken: We fixed that typo in the caption of Fig.1

  6. The index in Eq. (2) runs over all lattice sites and the other index in line 112 refers to sites of only one type of sublattice, with spin up or down. Action taken: We clarified the indexation after Eq. (2) and inside the footnote in line 122.

  7. Thanks for the suggestions. Action taken: We corrected the grammar mistakes accordingly.

  8. The ground state energy E_0 has been extracted from the mean-field Hamiltonian. However, it does not impact the dynamics. Consequently, we dropped it from the expressions after Eqs.(6). So, E_0 is the ground state energy. Action taken: We changed the definition for E_0, from “the constant energy” to “ the ground state energy”.

  9. We are thankful for this suggestion. It improved readability of the plot. Action taken: A vertical dashed line is plotted at the time point where the field is switched off and it is explained in the caption of the plot.

  10. We accept that we have used unclear notations there. Therefore we changed the equality for \vec{k}_0 at all places with proper 2d vectors. Action taken: We changed notations for the \vec{k}_0 mode at its all appearances as “\vec{k}_0=(0,0) and (\pi,\pi)”.

  11. We are sorry that this arrows are missing in the plot due to the missing replacement of the last version at last compilation Action taken: We replaced Fig.6 with the new version. There, two solid and dashed vertical black arrows are included to show the instant of switching.

  12. Thank you for your positive feedback. We appreciate the suggestion regarding the use of “the” and commas before main clauses beginning with “and”. We review the manuscript carefully and make the appropriate adjustments to improve clarity and readability. Action taken: We included “the” and commas at the appropriate places.

---

## Round 1 · Referee Report · Anonymous (Referee 2) · 2025-9-12

Strengths

A framework for analysis of switching processes in quantum antiferromagnets is suggested

Two different effects - dephasing and relaxation - are studied

Weaknesses

Mean-field Schwinger bosons, a simple version of Lindblad formalism for spin-lattice relaxation; what remains beyond these approximations?

Report

I think the paper meets the SciPost Physics acceptance criteria.

Recommendation

Ask for minor revision

  • validity: good
  • significance: good
  • originality: good
  • clarity: good
  • formatting: -
  • grammar: -

Author:  Asliddin Khudoyberdiev  on 2025-09-16  [id 5823]

(in reply to Report 2 on 2025-09-12)
Category:
remark
answer to question

We are glad that the message of the paper has become clear and grateful for your consideration of our work as suitable for publication in this journal. We also thank the Referee for highlighting the main point of the manuscript and the corresponding comments on the used theoretical tools. We will address them below.

Addressing the comments on Weakness:

To the best of our knowledge, time-dependent Schwinger mean-field theory is the only available quantum method to analyze and capture switching in quantum many-body systems. So far, only classical approaches were used based on Landau-Lifshitz-Gilbert equations [14,15]. Thus, we chose this approach and leave further methodological research to future work.
Note, that in Schwinger boson representation, the mixture or a superposition of the two boson flavors can represent any possible orientation of the magnetization. This is the real beauty and power of the representation, even though its mean-field treatment still represents an approximation. The use of this versatile approach for the description of switching magnetic order is a key element of our study. Another advantage of our approach is that it provides reasonable initial conditions because the quantum corrections in equilibrium are automatically included, at least on the mean-field level. The Schwinger mean-field results are very close to the numerically established values in equilibrium. The magnetization depends on all magnetic modes in the Brillouin zone so that dephasing is built-in. Therefore, we are confident that the obtained results are at least “quantitative” in the sense that they are based on unbiased computed numbers, not on qualitative, hand-waving arguments. The decay in magnetization oscillations due to dephasing of the modes is in-line with physical expectations and the effects of spin-lattice relaxation are captured within the established Lindblad formalism. Yet, we fully agree that future research should corroborate the quantitative reliability of the provided mean-field results by alternative methods.

In the present manuscript we used simple version of Lindblad formalism for an exemplary model to account for the spin-lattice relaxation. As it was highlighted in the conclusion of the manuscript, the singular behavior in our model requires further investigation, particularly in the context of more complex dissipation. For instance, the decay rate can acquire a momentum dependence or the relaxation can be chosen to conserve the total spin.

Action taken:
In the fifth paragraph of the introduction we highlighted the reason for the preferred method in our manuscript.

---

## Round 2 · Referee Report · Anonymous (Referee 2) · 2025-9-19

Weaknesses

As I wrote in my first report, the paper is good enough to be published in SciPost Physics. After revision, it becomes even better understandable. However, the authors did not react to my comment on the angle brackets. Well, I do not insist, but I am sure that the presentation will be more clear if the authors provide a formal definition of $\langle\ldots\rangle$ after Eqs. 5, 6 and then after Eq. 13 and 16, 17.

Report

I recommend to publish the paper.

Recommendation

Publish (meets expectations and criteria for this Journal)

---

## Round 2 · Referee Report · Anonymous (Referee 1) · 2025-9-19

Report

The authors have properly reacted to my comments. I suggest publication.

Recommendation

Publish (easily meets expectations and criteria for this Journal; among top 50%)

---

## Round 2 · List of Changes

We included additional explanations and references for read out techniques in antiferromagnets in the second paragraph of the introduction.

In the third paragraph of the introduction, we corrected the sentence in front of Ref. [18].

We fixed the typo in the caption of Fig.1.

In the fifth paragraph of the introduction we highlighted the reason for the preferred quantum approach in our manuscript.

We clarified the indexation after Eq. (2) and inside the footnote in line 122.

After Eqs.(6), we changed the definition for E_0, from “the constant energy” to “ the ground state energy”.

In Fig.2 (a), a vertical dashed line is plotted at the time point where the field is switched off and it is explained in the caption of the plot.

We included “initial state” and “final state” phrases in Fig.2 (b).

We changed notations for the \vec{k}_0 mode at its all appearances as “\vec{k}_0=(0,0) and (\pi,\pi)”.

At the end of fourth section, we provided numerical estimates for small 2d antiferromagnet and emphasized the possibility of the N\'eel vector tunneling only in small systems by citing Waldmann et al., Phys. Rev. Lett. 102, 157202 (2009).

In the third paragraph of the conclusion, we mentioned the possibility of switching at lower fields, by applying time-dependent form at the resonance frequency. Additionally, we commented on Landau-Zener-Stückelberg effect by citing [28].

We added the references 10, 11, 26, 27 and 28.

---

## Editorial Decision

published